# Computational imaging of moving objects obscured by a random corridor via speckle correlations

Tian Shi [1,2], Liangsheng Li [2✉], He Cai[2], Xianli Zhu[2], Qingfan Shi[1] & Ning Zheng [1✉]

Computational imaging makes it possible to reconstruct hidden objects through random media and around corners, which is of fundamental importance in various fields. Despite recent advances, computational imaging has not been studied in certain types of random scenarios, such as tortuous corridors filled with random media. We refer to this category of complex environment as a 'random corridor', and propose a reduced spatial- and ensemble-speckle intensity correlation (RSESIC) method to image a moving object obscured by a random corridor. Experimental results show that the method can reconstruct the image of a centimeter-sized hidden object with a sub-millimeter resolution by a low-cost digital camera. The imaging capability depends on three system parameters and can be characterized by the correlation fidelity (CF). Furthermore, the RSESIC method is able to recover the image of objects even for a single pixel containing the contribution of about $10^2$ speckle grains, which overcomes the theoretical limitation of traditional speckle imaging methods. Last but not least, when the power attenuation of speckle intensity leads to serious deterioration of CF, the image of hidden objects can still be reconstructed by the corrected intensity correlation.

[1] School of Physics, Beijing Institute of Technology, Beijing, China. [2] Science and Technology on Electromagnetic Scattering Laboratory, Beijing, China. ✉email: liliangshengbititp@163.com; ningzheng@bit.edu.cn

Relying on its ability to reconstruct an object hidden beyond the direct line of sight of a camera, computational imaging has recently attracted more and more interest[1–3]. The potential application of computational imaging ranges from robotic vision, autonomous driving, disaster relief, to medical imaging[1]. In the typical imaging scene widely studied previously, non-line-of-sight imaging always requires a relay surface located in the field of view of both the hidden object and the camera[3]. Non-line-of-sight methods, such as backprojection[4–7], light-cone transform[8], phasor field[9,10], computational periscope[11], and spatial coherence analysis[12,13], are able to reconstruct the images of objects obscured by corners, by using the spatiotemporal information of diffuse light reflected from the relay surface. New techniques and technology greatly improve the efficiency, economy, minimum resolution, and maximum working distance of non-line-of-sight imaging[8,9,11,14,15]. On the other hand, methods such as speckle correlations[16,17], wavefront-shaping[18], and speckle deconvolution[19] are capable of computational imaging through random media and around corners. In spite of recent advances, challenging but important computational imaging in another type of complex environment, 'random corridor' has received little attention. Random corridor represents a kind of tortuous corridor filled with random media, which often exists in rescue scenes such as dusty mines, smoke-filled ventilation ducts, and building corridors[20]. Imaging rescue targets obscured by a random corridor can greatly improve rescue efficiency and protect rescue crews.

In this paper, we propose a reduced spatial- and ensemble-speckle intensity correlation (RSESIC) method to image a moving object obscured by a random corridor. Experimental results show that this method can reconstruct a sub-millimeter resolution image of a centimeter-sized hidden object with a low-cost digital camera, whether the object is obscured by a random corridor with one or two corners. A correlation fidelity (CF) is defined to characterize the imaging capability of the system, and the dependence of the imaging capability on three system parameters: subspace side length, camera pixel count, and size ratio of pixel to speckle are investigated. If these parameters are inappropriately chosen, the CF will be poor and even unacceptable, resulting in computational imaging failure. The parameter range in which the RSESIC method can correctly image the hidden object is shown. The deterioration of CF caused by speckle intensity power attenuation is also studied. Based on the experimental and theoretical analysis, we demonstrate that the image of hidden objects can still be reconstructed after additional correlation renormalization.

## Results

**Imaging in a random corridor with one corner.** Figure 1a shows the schematic for an experimental imaging system. The hidden object is out of the camera's line of sight and obscured by a random corridor. The random corridor is L-shaped white plexiglass with one corner, as shown in the insets. The black dotted line represents an open boundary of the random corridor. The relay surface is not present for this scenario, and thus the laser incident on the hidden object can only propagate into the camera's field of view through the random corridor. The lens between the laser and the camera is used to defocus the beam to illuminate the entire object. The hidden object is a metal plate with three equally spaced slits mounted on an optical mobile platform (see Fig. 1e) and moves in the $XY$ plane. The position vector of the hidden object is denoted as $\mathbf{r}_o$. The displacement vector of the hidden object is denoted as $\Delta\boldsymbol{\rho}$, where $|\Delta\boldsymbol{\rho}| = \sqrt{\Delta x^2 + \Delta y^2}$. $\Delta x$ and $\Delta y$ are the projections of $\Delta\boldsymbol{\rho}$ on the X and Y axes,

respectively. A series of speckle intensity patterns that varies with $\Delta\boldsymbol{\rho}$ is recorded by a digital camera (Sony A6300).

We denote the raw speckle intensity images as $I_s(\mathbf{r}_c; \mathbf{r}_o + \Delta\boldsymbol{\rho})$, where $\mathbf{r}_c$ is the position vector on the sensor of the camera. Figure 1b shows $I_s(\mathbf{r}_c; \mathbf{r}_o + \Delta\boldsymbol{\rho})$ when a hidden object (see Fig. 1e) is located at $\mathbf{r}_o + \Delta\boldsymbol{\rho}$. $x_c$ and $z_c$ are projections of $\mathbf{r}_c$ on $X$ and $Z$ axes, respectively. The photographic parameters and image processing method are detailed in the "Methods" section. The raw speckle intensity image shown in Fig. 1b demonstrates that the transmitted light field illuminating the hidden object is completely randomized when it propagates into the camera's field of view, indicating that the hidden object cannot be directly imaged.

In order to decode the information carried by the speckle intensity images, a spatial speckle intensity correlation (SSIC) method was introduced in the past[21,22]. The method uses spatial average cross-correlation of $I_s(\mathbf{r}_c; \mathbf{r}_o + \Delta\boldsymbol{\rho})$ that varies with $\Delta\boldsymbol{\rho}$ to approximate the transmitted field autocorrelation of hidden objects $C_{ho}(\Delta\boldsymbol{\rho})$. When the object is hidden behind a thick random medium, the SSIC method accurately recovers $C_{ho}(\Delta\boldsymbol{\rho})$. Therefore, the image of the hidden object can be successfully reconstructed with $C_{ho}(\Delta\boldsymbol{\rho})$ and an iterative phase recovery algorithm[23]. However, when the object is obscured by a random corridor, the SSIC method fails to recover $C_{ho}(\Delta\boldsymbol{\rho})$ and reconstruct hidden object's image (experimental proof is provided in Supplementary Note 4).

Thus, we propose a RSESIC method to deal with the challenge of computational imaging hidden objects obscured by a random corridor. The RSESIC is defined as

$$C_{rd}(\Delta\boldsymbol{\rho}) = \left\langle \frac{[I_{rd}(\mathbf{r}_c; \mathbf{r}_o) - \bar{I}_{rd}][I_{rd}(\mathbf{r}_c; \mathbf{r}_o + \Delta\boldsymbol{\rho}) - \bar{I}_{rd}]}{\sigma_{rd}\sigma_{rd}} \right\rangle_{\mathbf{r}_c, \mathbf{r}_o} \quad (1)$$

where $I_{rd}$ is the subspace reduced speckle intensity, $\langle...\rangle_{\mathbf{r}_c}$ indicates spatial average and $\langle...\rangle_{\mathbf{r}_o}$ indicates ensemble average. Spatial average means the average over $\mathbf{r}_c$. Ensemble average means the average over $\mathbf{r}_o$ (more information is provided in the "Methods" section). $\bar{I}_{rd} = \langle I_{rd}(\mathbf{r}_c) \rangle_{\mathbf{r}_c}$ and $\sigma_{rd} = \sqrt{\langle I_{rd}^2(\mathbf{r}_c) \rangle_{\mathbf{r}_c} - \langle I_{rd}(\mathbf{r}_c) \rangle_{\mathbf{r}_c}^2}$ denote the mean and standard deviation of $I_{rd}$, respectively.

$I_{rd}$ is obtained through the subspace reduction process shown in Fig. 1b. First, the subspace standard deviation of each speckle intensity $I_s(\mathbf{r}_c; \mathbf{r}_o + \Delta\boldsymbol{\rho})$ is calculated using the following definition:

$$\sigma_s(\mathbf{r}_c; \mathbf{r}_o + \Delta\boldsymbol{\rho}) = \sqrt{\langle I_s^2(\mathbf{r}_c; \mathbf{r}_o + \Delta\boldsymbol{\rho}) \rangle_{sub} - \langle I_s(\mathbf{r}_c; \mathbf{r}_o + \Delta\boldsymbol{\rho}) \rangle_{sub}^2}$$

$$(2)$$

Here $\langle...\rangle_{sub}$ represents the average of the $I_s$ in a square subspace area on the camera sensor. The square subspace is centered at a given position vector $\mathbf{r}_c$. The side length of the square subspace is denoted as $l_{sub}$. For convenience, we further define a reduced subspace length $\hat{l}_{sub} = l_{sub}/l_{pixel}$, where $l_{pixel}$ represents the pixel side length of a camera. Then, $I_s$ at each position is reduced by its own standard deviation $\sigma_s$ to calculate the $I_{rd}$,

$$I_{rd}(\mathbf{r}_c; \mathbf{r}_o + \Delta\boldsymbol{\rho}) = \frac{I_s(\mathbf{r}_c; \mathbf{r}_o + \Delta\boldsymbol{\rho})}{\sigma_s(\mathbf{r}_c; \mathbf{r}_o + \Delta\boldsymbol{\rho})} \quad (3)$$

Before the subspace reduction, the standard deviations of $I_s$ at different positions are not equal. The $\sigma_s(\mathbf{r}_c; \mathbf{r}_o + \Delta\boldsymbol{\rho})$ plotted in Fig. 1b shows that the standard deviation of $I_s$ varies rapidly with spatial position. Therefore, $I_s(\mathbf{r}_c; \mathbf{r}_o + \Delta\boldsymbol{\rho})$ does not conform to Rayleigh statistics as a whole. The role of the subspace reduction is to obtain a reduced speckle intensity distribution $I_{rd}(\mathbf{r}_c; \mathbf{r}_o + \Delta\boldsymbol{\rho})$

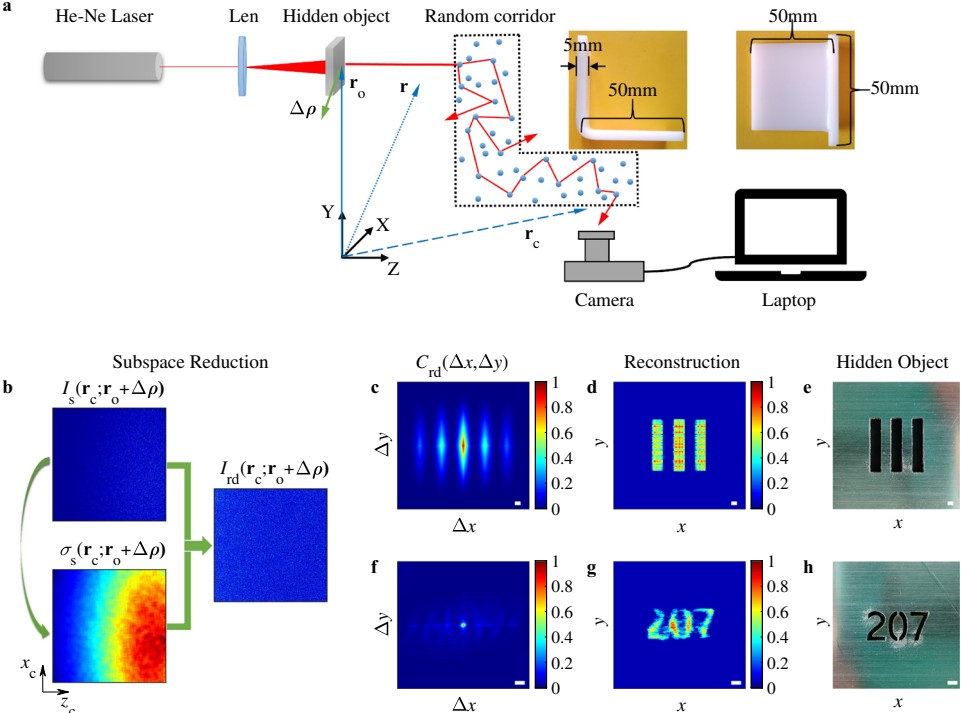

**Fig. 1 Imaging in a random corridor with one corner. a** Schematic of an experimental setup. **b** The subspace reduction process. **c** Cross-correlation of the $I_{rd}(\mathbf{r}_c; \mathbf{r}_o + \Delta\boldsymbol{\rho})$ in (**b**), clearly revealing distinctive patterns. **d** Object image reconstructed from the cross-correlation in (**c**) by a phase-retrieval algorithm. **e** A metal plate with three equally spaced slits. **f–h** As in (**c–e**) for a different object with a more complex shape. System parameters for imaging: $\hat{l}_{sub} = 50$, $R = 2.87$ and $N = 10^4$. Scale bar: 1 mm.

in which the standard deviation of light intensity at each point is almost equal. After the subspace reduction, $I_{rd}(\mathbf{r}_c; \mathbf{r}_o + \Delta\boldsymbol{\rho})$ agrees with the Rayleigh distribution as a whole. So the RSESIC method is insensitive to the spatial position of the observed speckle intensity pattern or the shape of random medium.

The common point between RSESIC method and previous SSIC method is that they are both speckle intensity cross-correlation methods. However, unlike the SSIC method, the RSESIC method calculates the cross-correlation of $I_{rd}$ and combines spatial and ensemble average. Subspace reduction enables RSESIC to image objects obscured by a random corridor. Ensemble averaging enables RSESIC to see through thicker random media.

In a random corridor with static random media, $C_{rd}(\Delta\boldsymbol{\rho})$ is approximately equivalent to $C_{ho}(\Delta\boldsymbol{\rho})$, which allows the RSESIC method to be used to image hidden objects (Supplementary Note 1). The $C_{rd}(\Delta\boldsymbol{\rho})$ for the hidden object in Fig. 1e is plotted, as shown in Fig. 1c, clearly revealing characteristic patterns. The resolution of the patterns is determined by the product of the moving speed of the hidden object $|\mathbf{v}|$ and the time interval of two photographs $\Delta\tau$. As mentioned before, the image of the hidden object can be reconstructed with $C_{rd}(\Delta\boldsymbol{\rho})$ and an iterative phase recovery algorithm[23,24] (more information is provided in the "Methods" section). The reconstructed image of the hidden object is plotted in Fig. 1d, which clearly demonstrates that the size and geometry of the hidden object are accurately restored. Similarly, Fig. 1f–h exhibit the imaging result of another hidden object, but with a complex shape. This further confirms the imaging competence of the RSESIC method for a random corridor with one corner.

For the RSESIC method, the consistency between $C_{rd}(\Delta\boldsymbol{\rho})$ and $C_{ho}(\Delta\boldsymbol{\rho})$, namely CF, determines the fidelity of the reconstructed image. The CF of the RSESIC method is dependent on three system parameters including reduced subspace length $\hat{l}_{sub}$, camera

pixel count $N$ and size ratio of pixel to speckle $R$. If these parameters are inappropriately chosen, the CF will be poor and even unacceptable, resulting in failure to image hidden objects. We explore the dependence of CF on $\hat{l}_{sub}$, $R$ and $N$ for the range of the parameters in which the RSESIC method can image the hidden object.

The size ratio of pixel to speckle on a camera is defined as

$$R = \frac{l_{pixel}}{d_{speckle}} \tag{4}$$

Here $d_{speckle}$ is the average diameter of the speckle grain on the camera sensor. $R$ determines the ability of the camera to resolve a single speckle grain. When $R \ll 1$, the morphology of the speckle grains can be clearly photographed by the camera. However, when $R \geq 1$, the contribution of multiple speckle grains will overlap in one pixel, so the morphology information of speckle grains will be lost completely. In our experiment, $l_{pixel} \approx 3.90\ \mu m$, $d_{speckle}$ is the product of the average diameter of the speckle grain on the surface of the white plexiglass and the overall magnification of the camera zoom lens. Theory and simulation show that the average diameter of the speckle grain on the surface of the random media is equal to a half wavelength[25,26]. The overall magnification of the camera zoom lens in the experiment is approximately equal to 4.31. Therefore, $d_{speckle} \approx 1.36\ \mu m$ and the minimum value of $R$ is 2.87.

We apply Eq. (1) to calculate $C_{rd}(\Delta x)$ for different parameters $\hat{l}_{sub}$, $R$ and $N$. The hidden object is a hollow metal plate shown in Fig. 1e and moves along the $X$ direction. Since the camera position and the focal length of the zoom lens (50 mm) are fixed, $R$ is increased by merging the pixels of the speckle intensity image. For example, $R$ increases from 2.87 to 8.61 when nine pixels are merged into one.

We first show the influence of $\hat{l}_{sub}$, $R$ and $N$ on the CF by directly comparing $C_{rd}(\Delta x)$ and $C_{ho}(\Delta x)$ in Fig. 2a, where $C_{ho}(\Delta x)$ is numerically calculated from the transmitted field of the hidden object (Supplementary Note 2). It can be found from Fig. 2a that the CF is proportional to $N$, and inversely proportional to $\hat{l}_{sub}$ and $R$. The decrease of $N$ gives rise to random fluctuations in $C_{rd}(\Delta x)$, and the increase of $\hat{l}_{sub}$ or $R$ causes positive background correlation in $C_{rd}(\Delta x)$. This suggests that the influence of $\hat{l}_{sub}$ and $R$ on $C_{rd}(\Delta x)$ are similar, but the effect from $N$ is different.

The Euclidean distance[27] is introduced to quantify the deviation between $C_{rd}(\Delta \boldsymbol{\rho})$ and $C_{ho}(\Delta \boldsymbol{\rho})$. Therefore, the CF of

the RSESIC method $d(C_{ho}, C_{rd})$ can be quantitatively calculated by the following definitions:

$$d(C_{ho}, C_{rd}) = 1 - \frac{\|\, C_{ho}(\Delta \rho) - C_{rd}(\Delta \rho)\,\|_2}{\|\, C_{ho}(\Delta \rho)\,\|_2 + \|\, C_{rd}(\Delta \rho)\,\|_2} \quad (5)$$

where $\|\ldots\|_2$ represents the Euclidean norm. When $C_{rd}(\Delta \rho) = C_{ho}(\Delta \rho)$, $d(C_{ho}, C_{rd}) = 1$. Otherwise, $0 < d(C_{ho}, C_{rd}) < 1$. The values of $d(C_{ho}, C_{rd})$ calculated by Eq. (5) are shown in the legend in Fig. 2a, which directly reflect the dependence of CF on $\hat{l}_{sub}$, $R$ and $N$. Since the curves of $C_{rd}(\Delta x)$ and $C_{ho}(\Delta x)$ almost overlap for $d(C_{ho}, C_{rd}) \geq 0.91$, it is reasonable to choose $d(C_{ho}, C_{rd}) = 0.91$ as a lower threshold of the validity of the RSESIC method.

The color maps in Fig. 2b shows $d(C_{ho}, C_{rd})$ for various $R$ and $N$ when $\hat{l}_{sub}$ is equal to 100 and 10, respectively, which further demonstrates the dependence of the CF on $\hat{l}_{sub}$, $R$ and $N$. The region surrounded by a contour line $d(C_{ho}, C_{rd}) = 0.91$ and two coordinate axes indicates the parameter range in which the RSESIC method can accurately recover $C_{ho}(\Delta x)$ and reconstruct the image of hidden objects. By comparing the two color maps, it can be found that reduction in $\hat{l}_{sub}$ significantly enlarges the optimal imaging region of $R$ and $N$. We also notice that the RSESIC method even holds for $R = 15$ when $\hat{l}_{sub} = 10$ and $N = 10^4$. This result breaks through the limit ($R < 1$) of previous speckle correlation methods[16,21,22], and provides insight into speckle-based imaging methods. In addition, when $\hat{l}_{sub} = 10$ and $R < 15$, $C_{ho}(\Delta x)$ can be accurately recovered with only $N = 10^3$ pixels, indicating that the imaging method is valid with a low-pixel camera.

**Imaging in a random corridor with two corners.** When an object is hidden behind several corners in a random corridor, the scattering of the corners and random media makes the light propagate from the hidden object through a very complex path to the observation point. In this case, it is extremely difficult to simulate the propagation of light. Besides, the power attenuation of the scattered light significantly enhances the influence of noise. Therefore, imaging objects hidden in such random corridors is very challenging. However, the RSESIC method does not need to solve the model for light propagation, and can eliminate the influence of ambient noise on the speckle intensity cross-correlation through extrapolation, thus resolving the barriers to imaging (Supplementary Note 3).

The schematic diagram to image objects hidden in a random corridor with two corners is shown in Fig. 3a. The hidden object is the metal plate with three slits, which moves in the $XY$ plane. The illustration shows the side and front view of the random corridor,

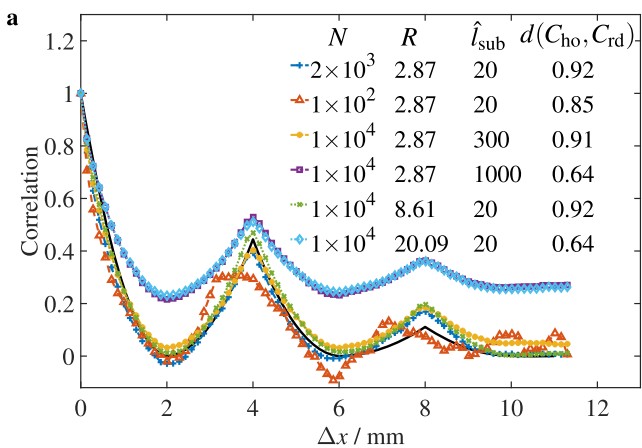

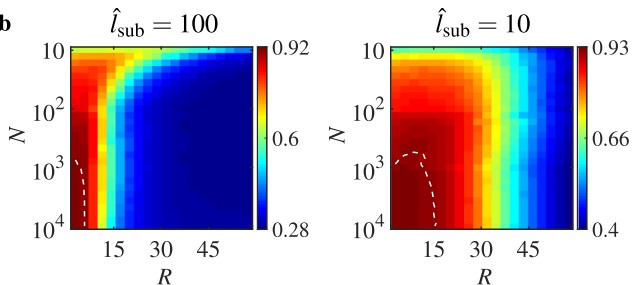

**Fig. 2 Influence of imaging system parameters on RSESIC method.** **a** Effects of $\hat{l}_{sub}$, $N$ and $R$ on $C_{rd}(\Delta x)$ are shown, and the black solid line denotes $C_{ho}(\Delta x)$. **b** Color maps of $d(C_{ho}, C_{rd})$ as a function of $R$ and $N$ for $\hat{l}_{sub} = 100$ and $= 10$, and the white dashed line is a contour line with $d(C_{ho}, C_{rd}) = 0.91$.

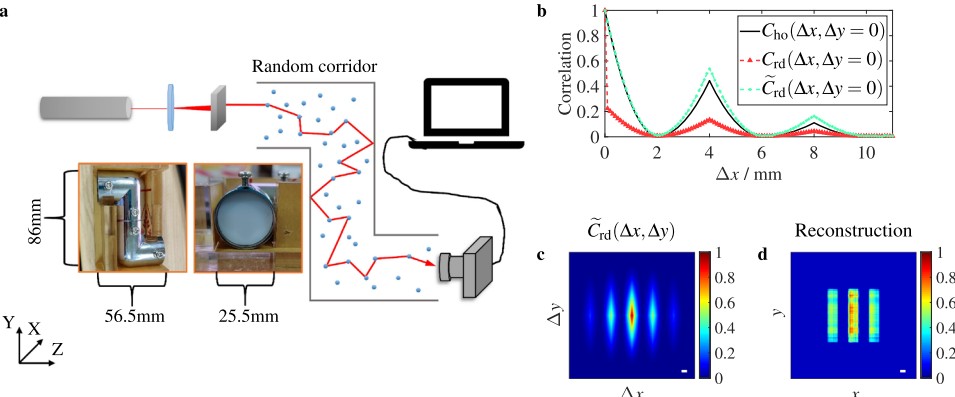

**Fig. 3 Imaging in a random corridor with two corners. a** Schematic of an experimental setup. **b** $C_{rd}$, normalized $\widetilde{C}_{rd}$ and $C_{ho}$ vs. $\Delta x$. $C_{ho}$ denotes the transmitted field autocorrelation of the hidden object. **c** Normalized $\widetilde{C}_{rd}(\Delta x, \Delta y)$. **d** Object image reconstructed from $\widetilde{C}_{rd}(\Delta x, \Delta y)$ by a phase-retrieval algorithm. System parameters for imaging: $\hat{l}_{sub} = 50$, $R = 2.87$ and $N = 10^4$. Scale bar: 1 mm.

which is a Z-shaped metal tube filled with white plexiglass. The first function of the metal tube is to increase the transmittance of incident light in the random corridor, so that the speckle intensity pattern behind the two corners can be detected by the camera. The second function of the metal tube is to enhance the scattering of incident light by the random corridor wall, so as to test the imaging performance of the RSESIC method under the coupling interference of the corridor wall and random medium.

We make a comparison between $C_{rd}(\Delta x)$ and $C_{ho}(\Delta x)$ in Fig. 3b. $C_{rd}(\Delta x)$ is calculated from the measured images of speckle intensity through the RSESIC method. Due to the influence of fast time-dependent noise, $C_{rd}(\Delta x)$ sharply drops at small displacement and significantly deviates from $C_{ho}(\Delta x)$, causing CF to decrease to $d(C_{ho}, C_{rd}) = 0.55$. Therefore, $C_{rd}$ cannot be directly used to reconstruct the image of the hidden object. However, the positions of peaks and valleys in $C_{rd}(\Delta x)$ and $C_{ho}(\Delta x)$ are identical, which implies that $C_{rd}$ still contains the information about $C_{ho}$. Fortunately, it is found that the adverse effect of ambient noise on the $C_{rd}$ can be eliminated by extrapolation and normalization, and then obtain a correlation $\widetilde{C}_{rd}$ approximately equivalent to $C_{ho}$. Here $C_{rd}(\Delta x, \Delta y)$ is normalized by $C_{rd}(0, 0)$ which is corrected by using extrapolation. The consistency between $\widetilde{C}_{rd}(\Delta x)$ and $C_{ho}(\Delta x)$ demonstrates that the RSESIC method can correctly restore the transmitted field autocorrelation of the hidden object behind two corners (see Fig. 3b). Then with the normalized $\widetilde{C}_{rd}(\Delta x, \Delta y)$ in Fig. 3c, the image of the hidden object was reconstructed by phase-retrieval, as shown in Fig. 3d. By comparing Fig.1e with Fig. 3d, it is clearly seen that the size and geometric characteristics of the hidden object are accurately restored, which confirms the ability of the RSESIC method to image in a random corridor with two corners.

In addition, the RSESIC method is further examined by imaging an object 'S' (see Fig. 4d) obscured by a random corridor with two corners. The area of 'S' is approximately equal to 30.79 mm$^2$, which is much smaller than that of the three equally spaced slits shown in Fig. 1e, 60 mm$^2$. Due to stronger attenuation of speckle intensity power, imaging of 'S' should be

more difficult. The imaging results demonstrate that the hidden object can still be imaged robustly, as shown in Fig. 4.

## Discussion

When an object is obscured by a random corridor, the difficulty of imaging the object is that the ballistic light is blocked and the relay surface is absent. We propose the RSESIC method to cope with this computational imaging challenge. Experimental results demonstrate that the method can reconstruct the image of the object obscured by a random corridor, using only speckle intensity images recorded by a digital camera. Applicable conditions for the method are summarized as follows: the random medium is static, the hidden object radiates or scatters coherent light, and motion information of the object must be known. The method is not limited to the transmitted speckle imaging system shown in this paper, but also applied to reflective laser speckle imaging systems and two-pass scattering systems (experimental proof is provided in Supplementary Notes 6 and 7).

Being different from most existing non-line-of-sight imaging methods, the RSESIC method can image hidden objects behind multiple corners without the spatiotemporal information of diffuse light reflected from a relay surface[3,8,9], and the reconstructed image is independent of the measurement position of multiple scattering light. Therefore, the method is complementary to the existing non-line-of-sight imaging methods and will promote further research on computational imaging in more complex scenes. In addition, the RSESIC method restores the image from the intensity cross-correlation of multiple scattering light rather than from solving a light propagation model, so the computation and complexity of the method are independent of the thickness of the random media and the number of corners. The ability of the RSESIC method to image objects behind multiple corners makes it promising to be applied to many fields such as disaster relief and automatic driving. If the moving distance of the hidden object and the measurement time of the camera are large enough, the RSESIC method can image a hidden object of any size. The actual imaging resolution of the method is determined by the product of the moving speed $|\mathbf{v}|$ of the hidden object and the time interval of two photographs $\Delta \tau$.

We analyze the effects of $\hat{l}_{sub}$, $R$ and $N$ on the CF of the RSESIC method and use a color map to point out an applicable parameter range for the method. The optimal combination of these three parameters can ensure the CF of the RSESIC method. For instance, with appropriate $\hat{l}_{sub}$ and $N$, the RSESIC method can accurately recover $C_{ho}$ and the image of the hidden object even for $R = 15$. This finding breaks through the limit ($R < 1$) of previous speckle correlation methods and proves that the speckle correlation method can image hidden objects when the speckle grains are not well resolved by a camera. Similarly, if $\hat{l}_{sub}$ and $R$ are appropriate, an average of $N = 10^3$ camera pixels is enough to make $C_{rd}$ and $C_{ho}$ match well. This result indicates that RSESIC method is suitable for low pixel cameras and has low requirements for storage and computing.

When the transmitted light of the hidden object is strongly attenuated during the propagation, ambient noise causes the $C_{rd}$ calculated by the RSESIC method to deviate significantly from $C_{ho}$. Therefore, the CF sharply deteriorates, resulting in imaging failure of hidden objects. Based on the experimental and theoretical analysis, we find that the mixture of correlation characteristic of fast time-varying ambient noise and the correlation of transmitted field of hidden objects leads to an abrupt decline in $C_{rd}$. We show that using an extrapolation method, the influence of noise in $C_{rd}$ can be reduced, and then the image of hidden objects can be reconstructed. This confirms the adaptability of the RSESIC method in a noisy environment. We further confirm that the adaptability of the RSESIC method in a noisy environment makes it possible to see through a random medium thicker than

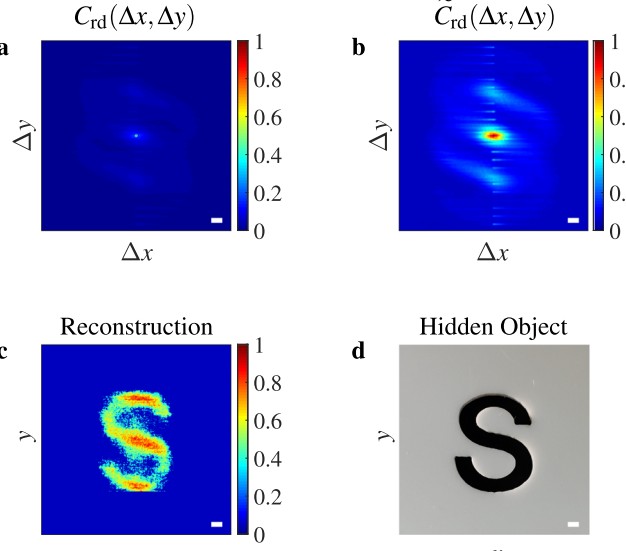

**Fig. 4 Imaging of a hidden object 'S' in a random corridor with two corners. a** Cross-correlation of the $I_{rd}(\mathbf{r}_c; \mathbf{r}_o + \Delta \rho)$ before renormalization. **b** Cross-correlation of the $I_{rd}(\mathbf{r}_c; \mathbf{r}_o + \Delta \rho)$ after renormalization. **c** Object image reconstructed from the cross-correlation in (**b**) by a phase-retrieval algorithm. **d** Photo of the hidden object. System parameters for imaging: $\hat{l}_{sub} = 50$, $R = 2.87$ and $N = 10^4$. Scale bar: 1mm.

that in the existing method (experimental proof is provided in Supplementary Note 5).

As a final note, the application of the RSESIC method is not only limited to imaging by laser, but also applicable for other types of coherent waves, such as acoustic waves, quantum waves and electromagnetic waves with multiple frequencies. Therefore, computational imaging with similar difficulties in other fields are expected to be addressed.

## Methods

**Experimental setup**. The experimental system consists of a 632.991 nm laser source (Thorlabs, HRS015B), a lens, a hidden object, a random corridor and a digital camera (Sony A6300). The zoom lens of the digital camera is set to a focal length of 50 mm. The distance between the zoom lens and random media is 1 mm. A speckle intensity pattern in an ~5.45 mm × 3.64 mm region of the white plexiglass surface is imaged on the 14-bit CMOS sensor of the camera. For imaging objects hidden behind one (two) corner(s), the shooting parameters of the camera are set to exposure time = 1/30 (1/10) s, aperture factor = 5.6, and ISO = 500 (6400), respectively.

**Image processing**. The photographs in RAW format were exported from the camera and further processed. Each photographic file is a 14-bit, 6024 × 4024-pixels raw image of which the color channels interleave according to the RGBG mode of a Bayer filter. Since the light emitted by the He–Ne laser is red, the data in the red channel of RAW files are extracted and used as the actual images of the speckle intensity. The size of speckle intensity image is 3012 × 2012-pixels.

**Spatial- and ensemble-speckle intensity correlation**. In Eq. (1), $\langle ...\rangle_{\mathbf{r}_c}$ is the average over the pixels of recorded speckle intensity images. Each speckle intensity image contains 3012 × 2012 intensity values, so the maximum spatial average times is 6,060,144 which is much larger than that required by the RSESIC method. From the definition of Eq. (1), it can be explicitly seen that the cross-correlation of speckle intensity reflects the similarity between a series of evolutionary speckle intensity images $I_s(\mathbf{r}_c; \mathbf{r}_o + \Delta\boldsymbol{\rho})$ for a moving hidden object and the reference speckle intensity image $I_s(\mathbf{r}_c; \mathbf{r}_o)$. When $\Delta\boldsymbol{\rho}$ is given, $m$ reference images of speckle intensity can be obtained by changing $\mathbf{r}_o$, and then $m$ correlation values can be calculated. $\langle ...\rangle_{\mathbf{r}_o}$ is the average of the $m$ correlation values. The intensity cross-correlations shown in Figs. 1, 3 and 4 are all calculated by $10^4$ spatial averages and 9 ensemble average. That is to say, the image of the hidden object is reconstructed from the data of speckle intensity measured on an area of 100 × 100-pixels of the CMOS sensor.

**Image reconstruction of hidden objects**. Autocorrelation of the transmitted field of a hidden object is defined as (Supplementary Note 1),

$$C_{ho}(\Delta\boldsymbol{\rho}) = \left| \frac{\int\int E_{ho}(x, y; \mathbf{r}_o)E_{ho}^*(x, y; \mathbf{r}_o + \Delta\boldsymbol{\rho})dxdy}{\int\int |E_{ho}(x, y; \mathbf{r}_o)|^2 dxdy} \right|^2$$

$$= \left| \int\int \hat{E}_{ho}(x, y; \mathbf{r}_o)\hat{E}_{ho}^*(x, y; \mathbf{r}_o + \Delta\boldsymbol{\rho})dxdy \right|^2$$

$$= \left| \mathscr{F}^{-1}\{|\mathscr{F}\{\hat{E}_{ho}(x, y)\}|^2\} \right|^2$$

Here $E_{ho}$ denotes the transmitted field of a hidden object, $\hat{E}_{ho}$ denotes the reduced dimensionless transmitted field, which is defined as,

$$\hat{E}_{ho}(x, y; \mathbf{r}_o + \Delta\boldsymbol{\rho}) = \frac{E_{ho}(x, y; \mathbf{r}_o + \Delta\boldsymbol{\rho})}{\sqrt{\int\int |E_{ho}(x, y; \mathbf{r}_o)|^2 dxdy}}$$

By the approximate relation $C_{rd}(\Delta\boldsymbol{\rho}) \approx C_{ho}(\Delta\boldsymbol{\rho})$, we have

$$|\hat{E}_{ho}(k_x, k_y)| = |\mathscr{F}\{\hat{E}_{ho}(x, y)\}| \approx |\sqrt{\mathscr{F}\{\sqrt{C_{rd}(\Delta\boldsymbol{\rho})}\}}|$$

The Fourier amplitude $|\hat{E}_{ho}(k_x, k_y)|$ of the reduced dimensionless transmitted field of a hidden object can be directly achieved. With substituting $|\hat{E}_{ho}(k_x, k_y)|$ and appropriate real-space support domain into the iterative phase recovery algorithm, $\hat{E}_{ho}(x, y)$ is reconstructed. Finally, $|\hat{E}_{ho}(x, y)|$ is the image of hidden object.

## Data availability

The data that support the findings of this study are available from the authors upon request.

## Code availability

The code that support the findings of this study are available from the authors upon request.

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

## Acknowledgements

We appreciate J.P. Johnson for a critical reading of this paper. This work was supported by the National Natural Science Foundation of China (NSF) (11974044) and National Key Research and Development Program of China (2016YFC1401001).

## Author contributions

L.L., N.Z., and Q.S. conceived the idea, T.S. and L.L. designed the experiments and performed numerical simulations. T.S., H.C., and X.Z. conducted the experiments and analyzed the data. T.S. and N.Z. wrote the paper.

## Competing interests

The authors declare no competing interests.
