## [Peer Review File · Nature Communications]

Computational imaging of moving objects obscured by a random corridor via speckle correlationsREVIEWER COMMENTS

Reviewer #1 (Remarks to the Author):

The manuscript describes a method to image a moving object with a known motion path through a scattering corridor. The method describes a new correlation based method called RSESIC that is claimed to enable this method. It seems to me that this reconstruction method is indeed new and relevant. The connection to NLOS imaging seems to be quite forced. When imaging through a turbid medium, the shape of the medium is indeed not relevant and no assumptions regarding shape are made in prior methods. It appears to me that prior methods like SSIC would have worked as well for the L shaped scattering medium that is introduced here. The method introduced by the author may be able to do a better job than prior methods in some way. Unfortunately, performance compared to prior methods is not evaluated.

1. Language and grammar could be improved in general.

2. I struggle to understand the rationale behind the problem setup. Imaging the object would be trivial if the laser was collimated onto the object. In that case the overall speckle intensity as a function of object position would reveal the object. Unfortunately the paper does not clearly state what the lens between the laser and the object does. I assume from the figure that it de-focuses the beam to illuminate the entire object. Please confirm.

3. This is of course not a realistic application scenario. As stated above, keeping the laser collimated would trivially lead to an image of the object without the need for reconstruction. What is the application scenario that the authors have in mind? I would assume that placing the laser next to the camera and looking at the object in reflection would be more relevant. This would probably require a pulsed laser and fast camera to eliminate direct back reflections. Placing a second scattering corridor between the laser and hidden object that is identical to the first one would be a simple way to implement a two pass scattering system with some application relevance.

4. The shape of the diffuser seems to be largely irrelevant. In essence the method is a speedup over previous work that should allow to see through thicker diffusers. If that is indeed achieved, it would be a relevant contribution. But the paper does not demonstrate that. Please provide a n experimental comparison or at least a realistic estimate of the performance of alternative methods.

5. Why is the prior SSIC not able to solve the reconstruction problem? It seems apart from SNR and speed and the required camera resolution it solves the same problem and should be able to achieve the same result. Was the failure of the SSIC method experimentally verified?

6. Please concisely summarize the differences between your method and the prior SSIC method.

7. The authors claim their method works with a lower resolution camera. Is that the only advantage of RSESIC over the prior art?

8. The method states that laser light is required. In fact the light only has to be monochromatic to create the necessary speckle. This could be achieved by filtering the light with a narrow band filter at the camera and would work for any light source as long as it is bright enough.

9. Some typos:

Line 15: *are* capable

line 32: reconstruct *a* sub-millimeter

line 171: when nine pixels *are* merged

In summary, the work includes two potential contributions:

- Showing that imaging through a thick turbid medium still works if the medium is bent in a

strange shape and the camera imaging the turbid medium doesn't look towards the scene. This seems obvious to me and I am not sure whether it is a significant contribution. It may be surprising to others.

- Introducing an improved algorithm for reconstruction through turbid medium. This is a potentially valuable contribution, but the manuscript does not contain any evaluation and comparison. As such there is not sufficient information to judge the method and important details that would be needed in a publication are missing.

I think the manuscript would need to be substantially revised with a detailed comparison to the prior art both in theory and experiment in order to be evaluated. Modifying the problem statement to something more like an actual NLOS experiment as suggested in (3) would also be an option.

Reviewer #2 (Remarks to the Author):

The authors present a method that allows to reconstruct images from light-fields that have propagate through complex-shaped random media (what they call 'random corridors').

The idea is very intriguing and definitely seems to be working quite well.

I also think that this work will inspire other ideas and further developments within the very broad field of imaging through complex or diffusive media.

The quality of the work is high and the results are convincing.

I would therefore recommend publication.

I do have just one question. I am, slightly puzzled by the definition of the reduced dimension speckle intensity given in equation 2. My intuitive understanding of the whole process is that the authors are taking the approaches that were previously developed by others and calculated the autocorrelation of speckle intensity patterns" however, these approaches worked only within the limits of the speckle memory effect. In the case of a random corridor, the speckle memory effect is going to be very limited or in other words, will be restricted to very small angles, i.e. a very small region on the sensor. So by analysing subspaces of the speckle pattern individually, they can recover some information that will be related to the small area over which the speckle memory is now effective. And then by averaging over all the subspaces within the image, they obtain good a good signal to noise ratio image.

But then I am puzzled as to why the $I_{rd} = I_s / \sigma$ is defined taking the full speckle image pattern I_s from the whole sensor, and only the σ is calculated over the reduced subspace. I would have expected that one would need to also confine the I_s to subspace regions too. Is there an intuitive explanation that the authors can add to their manuscript to try to properly explain the connection with previous work and the role of the subspaces so that one might possibly also intuitively understand equation 2?

Details aside, the method is obviously working quite well so the authors have correctly implemented the method. My doubt really is that I am not 100% I fully understand the details in a way that would allow to reproduce the results or, possibly, are there indeed slightly different ways of defining the reduced space intensities compared to equation 2 that might also work?

REVIEWER COMMENTS

Reviewer #1 (Remarks to the Author):

The manuscript describes a method to image a moving object with a known motion path through a scattering corridor. The method describes a new correlation-based method called RSESIC that is claimed to enable this method. It seems to me that this reconstruction method is indeed new and relevant. The connection to NLOS imaging seems to be quite forced. When imaging through a turbid medium, the shape of the medium is indeed not relevant and no assumptions regarding shape are made in prior methods. It appears to me that prior methods like SSIC would have worked as well for the L-shaped scattering medium that is introduced here. The method introduced by the author may be able to do a better job than prior methods in some way. Unfortunately, performance compared to prior methods is not evaluated.

1. Language and grammar could be improved in general.

Response: We thank the reviewer for this point. We invite a native speaker (a faculty in Brigham Young University-Idaho) to improve the language. Text changes are shown in red in the revised manuscript.

End of the response.

2. I struggle to understand the rationale behind the problem setup. Imaging the object would be trivial if the laser was collimated onto the object. In that case the overall speckle intensity as a function of object position would reveal the object. Unfortunately, the paper does not clearly state what the lens between the laser and the object does. I assume from the figure that it de-focuses the beam to illuminate the entire object. Please confirm.

Response: Thanks for the nice point. We performed several experiments to clarify the rationale, as shown in Fig.1. In these experiments, a laser beam is collimated on an object and scattered by the random medium behind the object. Photos of the scattered laser through the random medium, that is Fig.1b and d, are taken to demonstrate that the laser speckle intensity pattern behind the medium is a random distribution, which cannot directly image any geometric characteristics of the object. However, the information of the object can be extracted through the reconstruction of the RSESIC method. Therefore, the imaging method for hidden objects is significant.

The reviewer's speculation for the function of the lens is correct. The lens between the laser and the object is used to defocus the beam to illuminate the entire object, as shown in Fig. 1a and c. We

have added the following sentences to our revised manuscript, main text line 58 and 59:

“The lens between the laser and the camera is used to defocus the beam to illuminate the entire object.”

Figure 1. Experimental setup. A laser beam is collimated on an object, and then passes through a random medium. The photos of laser speckle are taken behind the random medium. a-b, An object in the shape of '207'. c-d, An object in the shape of 'III'

End of the response

3. This is of course not a realistic application scenario. As stated above, keeping the laser collimated would trivially lead to an image of the object without the need for reconstruction. What is the application scenario that the authors have in mind? I would assume that placing the laser next to the camera and looking at the object in reflection would be more relevant. This would probably require a pulsed laser and fast camera to eliminate direct back reflections. Placing a second scattering corridor between the laser and hidden object that is identical to the first one would be a simple way to implement a two-pass scattering system with some application relevance.

Response: We thank the reviewer for this nice point. As shown in Fig.1, even if the laser beam is collimated on the object, the laser speckle intensity pattern behind the random medium cannot directly reveal any geometric characteristics of the object. In this case, The RSESIC method can be used to reconstruct the hidden object.

In our mind, the application scenarios of the RSESIC method include the NLOS imaging shown in this paper, the reflective laser speckle imaging and the two-pass scattering system suggested by the

reviewer.

Figure 2. Imaging a hidden object with reflected laser speckle intensity images. a, Schematic diagram. b, Actual photo of the hidden object. c, Reflected laser speckle intensity image. d, Hidden object image reconstructed by the RSESIC method

Figure 3. Imaging an object hidden in a two-pass scattering system. a, Schematic diagram. b, Actual photo of the hidden object. c, Transmitted laser speckle intensity image. d, Hidden object

image reconstructed by the RSESIC method

Fig. 2 shows the experimental setup and results of imaging a hidden object with reflected laser speckle. Because there are no a pulsed laser and fast camera in our lab, the expanded laser directly shines on the hidden object in order to avoid the adverse effect of direct back reflections by a random medium, as shown in Fig. 2a. The RSESIC method can reconstruct the image of a hidden object by using a He-Ne laser and an ordinary camera. Fig. 2b shows the photo of the hidden object. The reflected laser speckle intensity image shown in Fig.2c confirms that the object is completely obscured by the random medium. Fig.2d demonstrates the hidden object image reconstructed by the RSESIC method. The good agreement between Fig. 2b and Fig. 2d shows that the RSESIC method is suitable for the reflection imaging system recommended by the reviewer.

Besides, Fig. 3 exhibits the imaging scenario and results for a two-pass scattering system. Fig. 3a shows the imaging scenario, in which the object is sandwiched between two opaque media. Fig. 3b shows the photo of the hidden object. The transmitted laser speckle intensity image shown in Fig.3c confirms that the object is completely obscured by the random media. Fig.3d demonstrates the hidden object image reconstructed by the RSESIC method. The reconstructed image (Fig. 3d) is consistent with the actual one of hidden object (Fig. 3b), showing that the RSESIC method is still valid for a two-pass scattering system suggested by the reviewer.

We have added the above two imaging scenes to the supplementary materials. Also, we have modified the conditions and scope of application of the RSESIC method in our revised manuscript, main text line 296-303,

“Applicable conditions for the method are summarized as follows: the random medium is static, the hidden object radiates or scatters coherent light, and motion information of the object must be known. The method is not limited to the NLOS system shown in this paper, but also applies to reflective laser speckle imaging systems and two-pass scattering systems (experimental proof is provided in Supplementary Note 6 and 7).”

End of the response

4. The shape of the diffuser seems to be largely irrelevant. In essence the method is a speedup over previous work that should allow to see through thicker diffusers. If that is indeed achieved, it would be a relevant contribution. But the paper does not demonstrate that. Please provide an experimental comparison or at least a realistic estimate of the performance of alternative methods.

Response: We are very grateful to the reviewer for the constructive advice. The RSESIC method, in a sense, is a speedup over previous work and allows to see through thicker random media. In order to confirm this, we performed experiments of imaging a hidden object behind random media with different thicknesses and compare the imaging quality.

Figure 4. Experimental comparison of imaging a hidden object obscured by random media with different thicknesses. a, Schematic diagram of the experimental system. b, Actual photograph of the hidden object. c, Photographs of random media with different thicknesses ($L=10, 30, 50\text{ mm}$). d, Comparison of reconstructed images between two methods, RSESIC and SSIC.

Fig. 4a is the schematic diagram of the imaging system, and the thickness of the random medium is denoted as L . Fig. 4b shows the hidden object that is a metal plate with three slits. Random media with thicknesses $L=10\text{mm}$, 30mm and 50mm were used in the experiment, as shown in Fig. 4c. In Fig. 4d, the hidden object images reconstructed by SSIC and RSESIC method are compared. It can be clearly seen in Fig. 4d that the SSIC method fails as $L>30\text{mm}$, but the RSESIC method can still

work when $L=50\text{mm}$. This result proves the advantage of the RSESIC imaging method for a thick random medium.

We have added the experiment of reconstructing hidden objects behind random media with different thicknesses to the supplementary material. Besides, we emphasize the advantages of RSESIC imaging method for thick random media in our revised manuscript, main text line 352-356,

“We further confirm that the adaptability of the RSESIC method in a noisy environment makes it possible to see through a random medium thicker than that in the existing method (experimental proof is provided in Supplementary Note 5).”

End of the response.

5. Why is the prior SSIC not able to solve the reconstruction problem? It seems apart from SNR and speed and the required camera resolution it solves the same problem and should be able to achieve the same result. Was the failure of the SSIC method experimentally verified?

Response: We thank the reviewer for the important question. The previous SSIC method does not apply the subspace reduction technology used in our manuscript, and directly uses the speckle intensity images $I_s(\mathbf{r}_c; \mathbf{r}_o + \Delta\boldsymbol{\rho})$ taken by a camera to reconstruct the hidden object, so it cannot solve the reconstruction problem in the case of NLOS. The failure of the SSIC method has been experimentally verified, as shown in Fig. 5a. Here, the image reconstructed by the SSIC method in the case of NLOS cannot reveal any features of the hidden object. By comparison, the RSESIC method successfully restores the image of hidden object from reduced speckle intensity images $I_{rd}(\mathbf{r}_c; \mathbf{r}_o + \Delta\boldsymbol{\rho})$, as shown in Fig. 5b.

We, therefore, added the above experimental results to the supplementary material.

Now reads: (main text line 88-91)

“However, the SSIC method fails to recover $C_{ho}(\Delta\boldsymbol{\rho})$ and reconstruct hidden objects under the NLOS conditions (experimental proof is provided in Supplementary Note 4).”

Figure 5. Comparison of reconstruction imaging between the RSESIC method and SSIC method under a NLOS condition. a, Raw speckle intensity images $I_s(\mathbf{r}_c; \mathbf{r}_o + \Delta\rho)$ and hidden object's image reconstructed by the SSIC method. b, Reduced speckle intensity images $I_{rd}(\mathbf{r}_c; \mathbf{r}_o + \Delta\rho)$ and hidden object's image reconstructed by the RSESIC method.

End of the response.

6. Please concisely summarize the differences between your method and the prior SSIC method.

Response: There are five main differences between our method and the previous SSIC method.

- 1) Our method calculates the cross-correlation of reduced speckle intensity, but the SSIC method calculates the cross-correlation of original speckle intensity.
- 2) Our method uses both spatial and ensemble averaging, while the SSIC method only uses spatial averaging.
- 3) The reconstructing image by our method does not depend on the observation position of speckle intensity pattern or the shape of random medium.
- 4) Our method allows imaging through a random medium thicker than that by SSIC method.
- 5) Our method does not need the morphological information of a single speckle grain, that is, it is particularly suitable for imaging with a camera with a low spatial resolution. However, the morphological information of a single speckle grain is necessary for the SSIC method.

End of the response.

7. *The authors claim their method works with a lower resolution camera. Is that the only advantage of RSESIC over the prior art?*

Response: In addition to imaging with a low-resolution camera, there are other two advantages in our method,

- 1) Our method is suitable for NLOS scenario which cannot be imaged by previous art.
 - 2) Our method allows imaging through a random medium thicker than that in the existing method.
- End of the response.

8. *The method states that laser light is required. In fact, the light only has to be monochromatic to create the necessary speckle. This could be achieved by filtering the light with a narrow band filter at the camera and would work for any light source as long as it is bright enough.*

Response: We appreciate the reviewer for this nice reminder. The reviewer is correct. The RSESIC method would work for any light source as long as it is bright enough. Speckle with a proper signal-to-noise ratio, rather than laser, is necessary for the RSESIC method. The reason for choosing laser is that laser has large power density.

End of the response.

9. *Some typos:*

*Line 15: *are* capable*

*line 32: reconstruct *a* sub-millimeter*

*line 171: when nine pixels *are* merged*

Response: The typos in the article have been corrected according to the reviewer's suggestions. Text changes are shown in red in the revised manuscript.

End of the response.

In summary, the work includes two potential contributions:

- Showing that imaging through a thick turbid medium still works if the medium is bent in a strange shape and the camera imaging the turbid medium doesn't look towards the scene. This seems obvious to me and I am not sure whether it is a significant contribution. It may be surprising to others.

- Introducing an improved algorithm for reconstruction through turbid medium. This is a potentially valuable contribution, but the manuscript does not contain any evaluation and comparison. As such there is not sufficient information to judge the method and important details that would be needed

in a publication are missing.

I think the manuscript would need to be substantially revised with a detailed comparison to the prior art both in theory and experiment in order to be evaluated. Modifying the problem statement to something more like an actual NLOS experiment as suggested in (3) would also be an option.

Response: According to the reviewer's suggestions, we supplement several experiments to show other applicable scenarios. Here, reflective laser speckle imaging system and two-pass scattering system can also be imaged by our method. Besides, we compare the performance of different imaging methods under various imaging conditions by experiments, and summarize the advantages of our imaging method. These detailed experimental comparisons between the RSESIC method and the existing technique are added in the supplementary materials.

End of the response.

Reviewer #2 (Remarks to the Author):

The authors present a method that allows to reconstruct images from light-fields that have propagate through complex-shaped random media (what they call 'random corridors'). The idea is very intriguing and definitely seems to be working quite well. I also think that this work will inspire other ideas and further developments within the very broad field of imaging through complex or diffusive media. The quality of the work is high and the results are convincing. I would therefore recommend publication.

I do have just one question. I am, slightly puzzled by the definition of the reduced dimension speckle intensity given in equation (2). My intuitive understanding of the whole process is that the authors are taking the approaches that were previously developed by others and calculated the autocorrelation of speckle intensity patterns, however, these approaches worked only within the limits of the speckle memory effect. In the case of a random corridor, the speckle memory effect is going to be very limited or in other words, will be restricted to very small angles, i.e. a very small region on the sensor. So by analysing subspaces of the speckle pattern individually, they can recover some information that will be related to the small area over which the speckle memory is now effective. And then by averaging over all the subspaces within the image, they obtain good a good signal to noise ratio image.

But then I am puzzled as to why the $I_{rd} = I_s / \sigma_s$ is defined taking the full speckle image pattern I_s from the whole sensor, and only the σ_s is calculated over the reduced subspace. I would have expected that one would need to also confine the I_s to subspace regions too. Is there an intuitive explanation that the authors can add to their manuscript to try to properly explain the connection with previous work and the role of the subspaces so that one might possibly also intuitively understand equation (2)?

Details aside, the method is obviously working quite well so the authors have correctly implemented the method. My doubt really is that I am not 100% fully understand the details in a way that would allow to reproduce the results or, possibly, are there indeed slightly different ways of defining the reduced space intensities compared to equation (2) that might also work?

Response: We thank the reviewer for the constructive point. In essence, the RSESIC method in this paper is a cross-correlation imaging method which reconstructs the image of hidden objects from multiple frame speckle intensity images, rather than an autocorrelation imaging method based on speckle memory effect. The complete imaging process of the RSESIC method is shown in Fig. 6. Firstly, a camera captures a series of speckle intensity images $I_s(\mathbf{r}_c; \mathbf{r}_o + \Delta\mathbf{p})$ that varies with the displacement of the hidden object $\Delta\mathbf{p}$, as shown in Fig. 6a. Secondly all speckle intensity images

are reduced according to the subspace reduction process shown in Fig. 6b. After this, a series of reduced speckle intensity images $I_{\text{rd}}(\mathbf{r}_c; \mathbf{r}_o + \Delta\boldsymbol{\rho})$ as a function of $\Delta\boldsymbol{\rho}$ is obtained, as shown in Fig. 6c. The following step is to calculate the cross-correlation of reduced speckle intensity image $I_{\text{rd}}(\mathbf{r}_c; \mathbf{r}_o + \Delta\boldsymbol{\rho})$ by using Eq (1) in the main text, where $C_{\text{rd}}(\Delta\boldsymbol{\rho})$ is defined as,

$$C_{\text{rd}}(\Delta\boldsymbol{\rho}) = \left\langle \frac{[I_{\text{rd}}(\mathbf{r}_c; \mathbf{r}_o) - \bar{I}_{\text{rd}}][I_{\text{rd}}(\mathbf{r}_c; \mathbf{r}_o + \Delta\boldsymbol{\rho}) - \bar{I}_{\text{rd}}]}{\sigma_{\text{rd}} \sigma_{\text{rd}}} \right\rangle_{\mathbf{r}_c, \mathbf{r}_o}$$

Fig. 6d and f show the cross-correlation of reduced speckle intensity calculated by this definition. Finally, the image of the hidden object is reconstructed from $C_{\text{rd}}(\Delta\boldsymbol{\rho})$ by using the iterative phase recovery algorithm. The reconstruction results are shown in Fig. 6e and g.

Figure 6. The whole imaging process of the RSESIC method is shown. a, Raw speckle intensity images $I_s(\mathbf{r}_c; \mathbf{r}_o + \Delta\boldsymbol{\rho})$ recorded by a camera. b, The subspace reduction. c, Reduced speckle intensity images $I_{\text{rd}}(\mathbf{r}_c; \mathbf{r}_o + \Delta\boldsymbol{\rho})$. d and f the cross-correlation of reduced speckle intensity images. e and g hidden object images reconstructed by the RSESIC method.

With Eq (3) in our revised manuscript, the reduced speckle intensity is defined as,

$$I_{\text{rd}}(\mathbf{r}_c; \mathbf{r}_o + \Delta\boldsymbol{\rho}) = \frac{I_s(\mathbf{r}_c; \mathbf{r}_o + \Delta\boldsymbol{\rho})}{\sigma_s(\mathbf{r}_c; \mathbf{r}_o + \Delta\boldsymbol{\rho})}$$

where $I_s(\mathbf{r}_c; \mathbf{r}_o + \Delta\boldsymbol{\rho})$, $\sigma_s(\mathbf{r}_c; \mathbf{r}_o + \Delta\boldsymbol{\rho})$ and $I_{\text{rd}}(\mathbf{r}_c; \mathbf{r}_o + \Delta\boldsymbol{\rho})$ are all defined in the whole space.

The purpose of the subspace reduction is to convert a speckle intensity distribution $I_s(\mathbf{r}_c; \mathbf{r}_o + \Delta\mathbf{p})$ into such a speckle distribution $I_{rd}(\mathbf{r}_c; \mathbf{r}_o + \Delta\mathbf{p})$ which has the same standard deviation of light intensity at each point. In the NLOS scenario, $I_s(\mathbf{r}_c; \mathbf{r}_o + \Delta\mathbf{p})$ denotes a random distribution of light intensity, but has independent standard deviation of light intensity at each point. So, the image of hidden objects cannot be successfully reconstructed by directly using $I_s(\mathbf{r}_c; \mathbf{r}_o + \Delta\mathbf{p})$ [see Fig. 5]. In contrast, $I_{rd}(\mathbf{r}_c; \mathbf{r}_o + \Delta\mathbf{p})$ has the same standard deviation of light intensity at all positions, so the image of hidden objects can be reconstructed from $I_{rd}(\mathbf{r}_c; \mathbf{r}_o + \Delta\mathbf{p})$ [see Fig. 6].

The subspace reduction is shown in Fig. 6b. Firstly, the subspace standard deviation $\sigma_s(\mathbf{r}_c; \mathbf{r}_o + \Delta\mathbf{p})$ of the observed speckle intensity image $I_s(\mathbf{r}_c; \mathbf{r}_o + \Delta\mathbf{p})$ is calculated by using Eq (2) in the main text,

$$\sigma_s(\mathbf{r}_c; \mathbf{r}_o + \Delta\mathbf{p}) = \sqrt{\langle I_s^2(\mathbf{r}_c; \mathbf{r}_o + \Delta\mathbf{p}) \rangle_{\text{sub}} - \langle I_s(\mathbf{r}_c; \mathbf{r}_o + \Delta\mathbf{p}) \rangle_{\text{sub}}^2}$$

Then, the light intensity of each position $I_s(\mathbf{r}_c; \mathbf{r}_o + \Delta\mathbf{p})$ is divided by its subspace standard deviation $\sigma_s(\mathbf{r}_c; \mathbf{r}_o + \Delta\mathbf{p})$ to obtain the subspace reduced speckle intensity $I_{rd}(\mathbf{r}_c; \mathbf{r}_o + \Delta\mathbf{p})$. $\sigma_s(\mathbf{r}_c; \mathbf{r}_o + \Delta\mathbf{p})$ can well approximate the standard deviation of light intensity $I_s(\mathbf{r}_c; \mathbf{r}_o + \Delta\mathbf{p})$ at each point, so all reduced light intensities $I_{rd}(\mathbf{r}_c; \mathbf{r}_o + \Delta\mathbf{p})$ own the same standard deviation.

The common point between the RSESIC and previous SSIC method is that they are both speckle intensity cross-correlation methods. The differences between them are summarized as below:

- 1) Our method (RSESIC) calculates the cross-correlation of reduced speckle intensity, but the SSIC method calculates the cross-correlation of original speckle intensity.
- 2) Our method uses both spatial and ensemble averaging, while the SSIC method only uses spatial averaging.
- 3) The reconstructing image by our method does not depend on the observation position of speckle intensity pattern or the shape of random medium.
- 4) Our method allows imaging through a random medium thicker than that by SSIC method.
- 5) Our method does not need the morphological information of a single speckle grain, that is, it is particularly suitable for imaging with a camera with a low spatial resolution. However, the morphological information of a single speckle grain is necessary for the SSIC method.

We introduce the subspace reduction method to reduce the original speckle light intensity, and thus

obtain the reduced speckle light intensity that is accord with Rayleigh distribution as a whole. The light intensity with Rayleigh distribution as a whole is the premise of cross-correlation reconstruction imaging. Although the original speckle intensity at every position follows Rayleigh distribution provided the scattering medium is disordered enough, the speckle light intensity as a whole does not match the Rayleigh distribution. It could be the simplest and reasonable way to make the speckle intensity pattern agree with Rayleigh distribution as a whole, with dividing light intensity of each position by its own standard deviation. So far, we have not found any other approaches to reduce the speckle intensity.

We have replaced Fig. 1b in main text by the subspace reduction process. Also, we have modified the contents between main text line 68-136 and added an explanation to properly demonstrate the role of subspace reduction and the connection with previous work.

Now reads:

“We denote the raw speckle intensity images as $I_s(\mathbf{r}_c; \mathbf{r}_o + \Delta\mathbf{p})$, where \mathbf{r}_c is the position vector on the sensor of the camera. Fig. 1b shows $I_s(\mathbf{r}_c; \mathbf{r}_o + \Delta\mathbf{p})$ when a hidden object [see Fig. 1e] is located at $\mathbf{r}_o + \Delta\mathbf{p}$. x_c and z_c are projections of \mathbf{r}_c on X and Z axes, respectively. The photographic parameters and image processing method are detailed in the **Methods** section. The raw speckle intensity image shown in Fig. 1b demonstrates that the transmitted light field illuminating the hidden object is completely randomized when it propagates into the camera's field of view, indicating that the hidden object cannot be directly imaged.

In order to decode the information carried by the speckle intensity images, a spatial speckle intensity correlation (SSIC) method was introduced in the past. The method uses spatial average cross-correlation of $I_s(\mathbf{r}_c; \mathbf{r}_o + \Delta\mathbf{p})$ that varies with $\Delta\mathbf{p}$ to approximate the transmitted field autocorrelation of hidden objects $C_{ho}(\Delta\mathbf{p})$. When the object is hidden behind a thick random medium, the SSIC method accurately recovers $C_{ho}(\Delta\mathbf{p})$. Therefore, the image of the hidden object can be successfully reconstructed with $C_{ho}(\Delta\mathbf{p})$ and an iterative phase recovery algorithm. However, the SSIC method failed to recover $C_{ho}(\Delta\mathbf{p})$ and reconstruct hidden objects under the NLOS conditions (experimental proof is provided in Supplementary Note 4).

.....

I_{rd} is obtained through the subspace reduction process shown in Fig. 1b. First, the subspace standard deviation of each speckle intensity $I_s(\mathbf{r}_c; \mathbf{r}_o + \Delta\mathbf{p})$ is calculated using the following definition,

$$\sigma_s(\mathbf{r}_c; \mathbf{r}_o + \Delta\mathbf{p}) = \sqrt{\langle I_s^2(\mathbf{r}_c; \mathbf{r}_o + \Delta\mathbf{p}) \rangle_{\text{sub}} - \langle I_s(\mathbf{r}_c; \mathbf{r}_o + \Delta\mathbf{p}) \rangle_{\text{sub}}^2} \quad (2)$$

Here $\langle \dots \rangle_{\text{sub}}$ represents the average of the I_s in a square subspace area on the camera sensor. The square subspace is centered at a given position vector \mathbf{r}_c . The side length of the square subspace

is denoted as l_{sub} . For convenience, we further define a reduced subspace length $\hat{l}_{\text{sub}} = l_{\text{sub}} / l_{\text{pixel}}$, where l_{pixel} represents the pixel side length of a camera. Then, I_s at each position is reduced by its own standard deviation σ_s to calculate the I_{rd} ,

$$I_{\text{rd}}(\mathbf{r}_c; \mathbf{r}_o + \Delta\boldsymbol{\rho}) = \frac{I_s(\mathbf{r}_c; \mathbf{r}_o + \Delta\boldsymbol{\rho})}{\sigma_s(\mathbf{r}_c; \mathbf{r}_o + \Delta\boldsymbol{\rho})} \quad (3)$$

Before the subspace reduction, the standard deviations of I_s at different positions are not equal. The $\sigma_s(\mathbf{r}_c; \mathbf{r}_o + \Delta\boldsymbol{\rho})$ plotted in Fig. 1b shows that the standard deviation of I_s varies rapidly with spatial position. Therefore, $I_s(\mathbf{r}_c; \mathbf{r}_o + \Delta\boldsymbol{\rho})$ does not conform to Rayleigh statistics as a whole. The role of the subspace reduction is to obtain a reduced speckle intensity distribution $I_{\text{rd}}(\mathbf{r}_c; \mathbf{r}_o + \Delta\boldsymbol{\rho})$ in which the standard deviation of light intensity at each point is almost equal. After the subspace reduction, $I_{\text{rd}}(\mathbf{r}_c; \mathbf{r}_o + \Delta\boldsymbol{\rho})$ agrees with the Rayleigh distribution as a whole. So the RSESIC method is insensitive to the spatial position of the observed speckle intensity pattern or the shape of random medium.

The common point between RSESIC method and previous SSIC method is that they are both speckle intensity cross-correlation methods. However, unlike the SSIC method, the RSESIC method calculates the cross-correlation of I_{rd} and combines spatial and ensemble average. Subspace reduction enables RSESIC to image in NLOS conditions. Ensemble averaging enables RSESIC to see through thicker random media.”

End of the response.

REVIEWERS' COMMENTS

Reviewer #1 (Remarks to the Author):

All my technical concerns have been addressed.

I still think, this method is fundamentally a method for imaging through a turbid medium and the experiments done fit that problem statement better than NLOS imaging.

I recommend changing the title, narrative, and contribution statement accordingly.

Reviewer #2 (Remarks to the Author):

The authors have replied to my queries and I would now support publication of the work without the need for further revision.

REVIEWER COMMENTS

Reviewer #1 (Remarks to the Author):

All my technical concerns have been addressed. I still think, this method is fundamentally a method for imaging through a turbid medium and the experiments done fit that problem statement better than NLOS imaging. I recommend changing the title, narrative, and contribution statement accordingly.

Response: We appreciate the reviewer for the constructive advice. In this paper, we propose a reduced spatial- and ensemble-speckle intensity correlation method to image a moving object obscured by a random corridor. Therefore, we have revised the title of the manuscript as follows,

“Computational imaging of moving objects obscured by a random corridor via speckle correlations”.

Besides, we have changed inappropriate ‘non-line-of-sight (NLOS)’ expressions in the context, and revised the narrative and contribution statement accordingly. The first sentence in the second paragraph in the section of Introduction reads,

“In this paper, we propose a reduced spatial- and ensemble-speckle intensity correlation (RSESIC) method to image a moving object obscured by a random corridor.”

The second sentence in the first paragraph in the section of Discussion reads,

“We propose the RSESIC method to cope with this computational imaging challenge.”

The fifth sentence in the first paragraph in the section of Discussion reads,

“The method is not limited to the transmitted speckle imaging system shown in this paper, but also applied to reflective laser speckle imaging systems and two-pass scattering systems.”

All the text changes are highlighted in red in the revised manuscript.

End of the response.

Reviewer #2 (Remarks to the Author):

The authors have replied to my queries and I would now support publication of the work without the need for further revision.